# Prescribing errors in a Brazilian teaching hospital: Causes and underlying factors from the perspective of junior doctors

**Gislaine F. Bonella**[1,2]*, **Letícia da S. Alves**[1], **Alicia R. N. D. de Souza**[3], **Carlos H. M. da Silva**[1]

1 Graduate Program in Health Sciences, Faculty of Medicine, Federal University of Uberlândia, Uberlândia, MG, Brazil, 2 Hospital Pharmacy Department, University Hospital, Federal University of Uberlândia, Uberlândia, MG, Brazil, 3 Department of Psychiatry and Legal Medicine, Faculty of Medicine, Federal University of Rio de Janeiro, Rio de Janeiro, RJ, Brazil

* gibonella@gmail.com

**Data Availability Statement:** All relevant data are within the paper and its Supporting Information files.

**Funding:** The authors received no specific funding for this work.

## Abstract

### Introduction

In university hospital settings most prescriptions are written by junior doctors, who are more likely to make prescribing errors than experienced doctors. Prescribing errors can cause serious harm to patients and drug harm differs among low, middle and high-income countries. In Brazil, few studies have investigated the causes of these errors. Our aim was to explore medication prescribing errors in a teaching hospital, their causes, and underlying factors from the perspective of junior doctors.

### Method

Qualitative, descriptive and exploratory study that used a semi-structured individual interview with questions related to the planning and execution of prescriptions. It was conducted with 34 junior doctors who graduated from twelve different universities located in six Brazilian states. The data were analyzed according to the Reason's Accident Causation model.

### Results

Among the 105 errors reported, medication omission stood out. Most errors resulted from unsafe acts during execution, followed by mistakes and violations. Many errors reached the patients; unsafe acts of rule violations and slips accounted for the majority. Work overload and time pressure were the most frequently reported causes. Difficulties faced by the National Health System and organizational problems were identified as latent conditions.

### Conclusion

The results reaffirm international findings about the severity of prescribing errors and the multifactorial aspect of their causes. Unlike other studies, we found a large number of violations, which, from the interviewees' perspectives, are related to socioeconomic and cultural patterns. The violations were not seen or mentioned by the interviewees as violations, but

**Competing interests:** The authors have declared that no competing interests exist.

as difficulties in accomplishing their tasks on time. Knowing these patterns and perspectives is important for implementing strategies to improve the safety of both patients and professionals involved in the medication process. It is suggested that the exploitation culture of junior doctors' work be discouraged and that their training be improved and prioritized.

## Introduction

As with all types of medication errors, prescribing errors are common in hospitals [1, 2]. They may occur in 50% of all hospital admissions [3], 7% to 8.9% of manual medical prescriptions [2, 4] and 2% to 7% of electronic prescriptions [5–7]. Serious harms caused by prescribing errors are the most frequent [8], and may increase hospitalization times, healthcare costs and mortality rates [9]. The scale and nature of drug harm in health systems differ among low, middle and high-income countries [10].

Prescribing errors are pointed as a major concern in published studies on some hospitals in Brazil, as they occur in 43.5% to 44.5% of handwritten, mixed and pre-typed prescriptions [11, 12] and in 8.0% of electronic prescriptions [13]. Inadequate prescriptions of antimicrobials for prophylaxis [14] and infection treatments [15, 16] have also been identified.

Junior doctors are more likely to make prescribing errors than more experienced doctors [1] and usually account for most errors in teaching hospitals [17]. International studies on the prevalence of prescribing errors by junior doctors in the United States of America (USA) and in the United Kingdom report errors in 5.9% to 7.5% of prescribed items [18, 19]. Ferracini et al. [20] published data on a hospital in Brazil pointing at a prescribing error rate of 15%.

To better understand prescribing errors, studies have used "Reason's Accident Causation model" [21] as it addresses behavioral factors linked to errors, enables integrating individual and systemic approaches, and is widely used in hospital contexts [3, 4, 17, 18, 22–29].

The model classifies active failures into errors and rule violations. Errors are divided into two classes: execution errors and planning errors. Execution errors are classified into slips and lapses. Planning errors, called mistakes, are classified into knowledge-based mistakes (KBMs) and rule-based mistakes (RBMs).

The model also points out error-producing conditions which lead the prescriber to commit errors, latent conditions related to organizational processes and management, and failures in prevention barriers. Further explanation on the classification and definitions used in Reason's Accident Causation model are given in Table 1.

Studies point the multifactorial causes of prescribing errors, such as lack of training and inexperience, fatigue, stress, work overload, poor communication between health professionals, reluctance to question superiors' decisions and inadequate availability of training [22].

The problem of prescription errors seen in an individual manner is associated, in part, with human behavior and its failures. This, in turn, is related to a cognitive and emotional structure not easily understood by quantitative research alone. Hence, we chose a qualitative approach to investigate subjective aspects related to errors and an analysis framework based on Reason's Accident Causation model [21].

Understanding the underlying causes related to errors is essential for a better comprehension of the problem and important to design educational strategies and organizational systems towards error reduction and patient safety [30].

In Brazil no qualitative study carrying out an in-depth investigation of the underlying causes of prescribing errors from the perspective of junior doctors has been found. This research is made necessary as it will be useful for clinicians, mainly physicians, nurses, and

**Table 1. Definitions used in Reason's Accident Causation model" [21].**

| Classification | Definition |
|---|---|
| **1. Active failures/ unsafe acts** | Unsafe acts performed by who are in direct contact with the patient or system. They are at the sharp end of the error. They are divided into "errors and violations". |
| **1.1 Errors** | Are non-voluntary /active failures and are classified into: planning failures and execution failures. |
| **Planning Failure** | Those resulting from the correct execution of an inappropriate or incorrect plan. |
| a) Knowledge-based mistake | Happen due to lack of knowledge or inexperience in medication prescription and refer to the incorrect choice of a plan, such as wrong medication prescription or dose. |
| b) Ruled based mistake | Inappropriate choice of plan to achieve a goal. They are likely to happen due to the incorrect use of knowledge in the decision-making context, for example, prescribing the dose for standard clinical conditions without considering adjustments for specific conditions such as renal function and patient age; cases where it is believed the decision at the moment is correct. |
| **Execution Failure** | Those deriving from failure to execute a correct plan |
| a) Slips | An error that is caused by a failure in performing an intended action and that is replaced by another action, for example, failures to recognize and select drugs with similar names in the electronic prescription system. |
| b) Lapses | Memory or attention faults, like planning to interrupt medication but not doing it. |
| **1.2 Violation** | Voluntary or intentional active failures and consist of deliberately breaking codes of conducts and routines. They are voluntary actions in which rules are deliberately ignored, for example, prescribing without a proper assessment of the patient's current clinical conditions. |
| **2. Error-producing conditions** | Are conditions that predispose an individual to making an error. These also termed contributory factors and can relate to factors in the environment (e.g. work overload), the task (e.g. failure to check prescription) the team (e.g. absent or inadequate preceptorship), individuals (e.g. lack of knowledge) or the patient (e.g. complexity). |
| **3. Latent conditions** | Are related to organizational processes and management, and failures in prevention barriers They are not a direct cause of errors but can translate into "error-producing conditions" and leads to weaknesses in the defence barriers allowing errors to manifest. |

The examples are taken from the reports of the participants in this study.

pharmacists, whose professions are directly linked to medication prescription errors, and also contribute to the reducing the risk of harm to patients.

Although other international studies have addressed this perspective, the same type of investigation in a different and not yet properly explored environment will, on one hand, contribute to a better global understanding of the problem and, on the other, will relate to problems and specific socio-cultural reality of developing countries, where interventions are necessary and sought after.

The aim of the study was to explore medication prescribing errors in a teaching hospital, their causes, and underlying factors from the perspective of junior doctors.

# Method

## Qualitative approach and research paradigm

We conducted a qualitative, descriptive exploratory study, collecting data through individual interviews. Focusing on the subjective and personal dimension of human error, we adopted a constructivist paradigm, assuming that there are multiple and distinct subjective realities (perspectives) [31]. Within this paradigm, we adopted Reason's Accident Causation model [21] for the concept-based exploration of themes. The epistemic dimension was phenomenological [31], starting from the way reality appears to the participants themselves, taking them as autonomous subjects of medical practice.

## Researcher characteristics and reflexivity

The author who conducted all the interviews worked at the hospital pharmacy at the study site but was not known or acquainted with the participants before starting the study. The medical prescriptions are usually printed directly in the pharmacy sector and when a problem is detected, the pharmaceutical team contacts the doctor by phone for clarification. No member of the study taught, supervised, or had any authority over participants in the study.

The interviewer was fully aware of her potential assumptions during the interviews, taking care that these did not interfere with the accounts, she sought not to deviate from the questions contained in the interview guide, ensured that all data could be reviewed and discussed by the entire research team and that disagreements could be discussed and resolved by consensus ensuring neutrality and objectivity to the study. Reasons and interests were scientific in order to understand the underlying aspects of prescribing errors and were presented to the participant before starting the interview.

## Context

The study was carried out in a teaching hospital and service provider of the Brazilian Unified Public Health System (*Sistema Único de Saúde*—SUS) containing 530 hospital beds and served by a computerized prescription system. This system does not allow the locking of the prescribed medications, but it does have a maximum dose alert for some types of medications, such as high-risk medications.

In our study, the term "junior doctor" refers to the doctors who, after finishing their undergraduate degree, go on a residency program to specialize in an area of medicine. The vast majority are newly graduated doctors.

## Sampling strategy

The sampling universe consisted of doctors from the residency program who were training at the hospital in the period between April and July 2018, when the interview was performed. First, second and third-year residents were included in the study, those with over three years and those on vacation or on sick leave during the study period were excluded.

Invitation to participate in the study was initially done through an email to all (632) junior doctors enrolled in the Medical Residency Program, however there were no responses. The sector responsible for the medical residency program made available a list with the junior doctors and their specialties. Thus, 40 participants were individually recruited in their workplace, either in person or by telephone. As one of the aims of the study was to identify contextual causes and previous errors in different hospital environments, junior doctors of both genders and from different settings (wards, adult and pediatric intensive care units and emergency care units) were selected. The study also sought to contemplate several medical specialties. None of the recruited participants refused to participate in the study, however, six did not participate due to their impossibility to schedule appointments with the interviewer, which, according to them, was due to their residency schedule and work overload. Theoretical saturation was obtained with 34 interviews [32].

## Ethical issues

The study, recruitment and consent were approved without restrictions by the Human Research Ethics Committee of the Federal University of Uberlandia on 03/28/2018, under process number 2.570.103. All participants gave their written informed consent before participating. Ethical risks were minimized by adopting total confidentiality regarding the data and

identity of the participants (anonymity through impersonal coding), and through rigorous care when explaining and obtaining participants' consent. The study participant was informed through the written consent form and the interviewer that their participation was voluntary, had no financial expense or gain and that they were free to opt out at any time, without any harm or coercion. The interviews were conducted in a private place, out of working hours and the audio was recorded. After being transcribed the audio files were deleted.

## Data collection, instrument and processing

An interview topic guide was used with questions related to the "planning" and "execution" of the prescription (S1 and S2 Files), and was an adaptation of the one used in Lewis et al. [26], with due authorization. This approach provided a detailed account of prescribing decisions (specific behaviors when planning and executing the prescription) [26].

The interviews were conducted by the first pharmaceutical MD author G, as she has over 20 years of experience in hospital pharmacy, routinely dealing with prescription errors, at a time chosen by the participant and respecting his or her availability. The interviews lasted between 15 and 35 minutes and were audio-recorded and transcribed in full. Field notes were taken, however interviews were not repeated and transcripts were not returned to the participant for comment or correction."

In the interview the participant was asked to report prescribing errors he/she had committed and/or errors detected in prescriptions by other junior doctors throughout their entire professional life. The definition of error used and shared with participants was "when, as a result of a prescribing decision or prescription writing process, there is an unintentional, significant reduction in the probability of treatment being timely and effective or increase in the risk of harm when compared with generally accepted practice" [33].

## Data analysis

For a qualitative investigation exploring subjective perspectives we used a Framework Approach [34] for data analysis. A period of familiarization with the raw data (listening to the recordings, reading the transcripts, checking notes with reflective care) was previously undertaken by G and L to ensure that they had relevant information for the study's purpose. Reason's Accident Causation model [21] was used to categorize and present the data. The development of the analytic framework and a categorization and coding according to "Reason's Accident Causation model" were carefully elaborated by G. This model was the most commonly used theoretical model when considering prescribing errors [3, 4, 17, 18, 22–29]. To ensure higher accuracy, the application of this framework and the mapping of the data were performed independently by G and L and the differences were settled by consensus. Tables and organizational charts were constructed to facilitate understanding and interpretation.

The computer software program NVivo© [35] was used to assist in the organization of the data. Codes and sub-codes to qualify the 34 cases were previously defined and used during the analysis process to identify potential themes. Prescribing errors were then counted and named according to the taxonomy in Otero-López et al. [36].

## Rigor of data analysis

We sought to link the data analysis to theoretical constructs, theories used in preliminary work, and codes derived from the theoretical framework of relevant literature to ensure greater reliability and credibility. National and international recommendations/protocols on good prescription practice and patient safety [37–40] were used as objective aspects to guide our

interpretation and critical discussion of the data to avoid evaluative bias. To ensure researcher neutrality and objectivity, all data was made available for approval by all the authors, two of whom were senior doctors and experienced researchers in qualitative research (A) and health sciences (C). Once the findings were sufficiently clarified with the chosen structural analysis, the interaction between findings and theories enabled transferability. The relevant findings and interpretations present applicability beyond the greater understanding of the problem, such as encouraging new research (in Brazil and in other countries), promoting critical-reflective evaluation and suggesting improvements in care, benefits for the participants and improvements in patient safety.

## Results and analyses

A total of 34 junior doctors graduated from 12 medical schools in six Brazilian states participated in the study and reported 105 prescribing errors. The main type of prescribing error reported was omission of medication [36].

For a better qualitative understanding of the data, literal reports were selected to illustrate the main results. The quotes were referenced using numbers representing the participant (P) code, eg., P 01, P02, etc.

Most errors reported were caused by slips, followed by violations, knowledge-based mistakes, lapse and rule-based mistakes (Fig 1).

## Execution errors: Slips and lapses

Most slips were related to medication omission and wrong dose, wrong medication, wrong route of administration and wrong patient. Due to name similarity between the drugs and

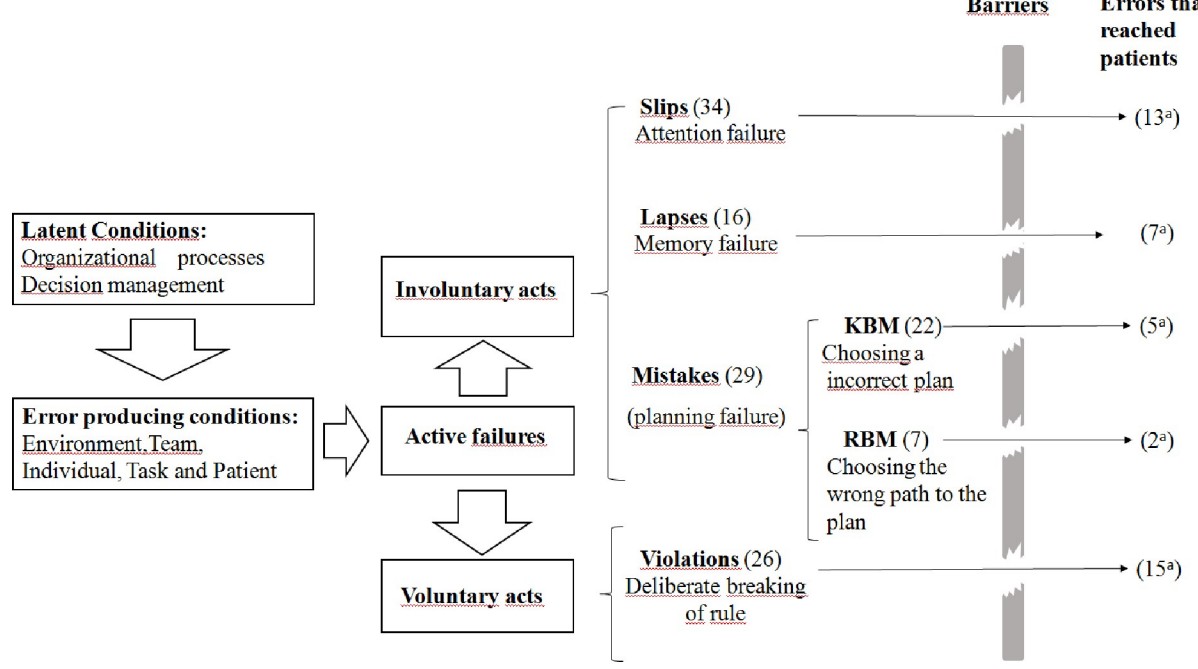

**Fig 1. Active failures, error-producing and latent conditions (adapted from the Reason's Accident Causation model)** [21]. KBM = knowledge-based mistake. RBM = Rule-based mistake. Numbers = number of errors. [a] = number of errors reaching patients. The active failures/unsafe acts were identified from the report of the participants of this research. Note: the gaps in the barriers represent failures in the system. Source: the researcher.

their alphabetical proximity in the prescribing electronic system, the interviewee selected the wrong medication.

> *"The drugs had similar names, nitroglycerin and nitroprusside, vasodilators, I know the names have to be in alphabetical order, but this led me to making this mistake."* (P03)

Most lapses consisted in medication omission or unnecessary drug prescription. Difficulty in remembering to suspend unnecessary medications and re-prescribe temporarily suspended medications were examples of lapses reported. One of the interviewees forgot to suspend the thrombosis prophylaxis in the preoperative state and the surgery had to be canceled.

> *"I've done it twice, and it makes me upset. Once we had to cancel the surgery and the other time we managed to revert the situation to allow time to operate."* (P19)

## Planning errors: Mistakes

Most mistakes involved KBM related to lack of knowledge and inexperience with the computer system and lack of knowledge in relation to the proper planning of prescriptions regarding dose, route of administration, interval of administration, dilution, diluent, posology and instructions for use. Due to lack of knowledge, one of the interviewees had to rewrite the prescription for magnesium sulfate to a patient with preeclampsia.

> *"When I started here we didn't know much about dosage, so there were times when I had to make out more than one prescription for a medication due to the dilution, the saline solution, the administration, especially when it was magnesium sulfate, because of preeclampsia at GO [Gynecology and Obstetrics]. We're a little inexperienced when it comes to dilution."* (P14).

Some respondents were unaware that instructions for use should be prescribed by doctors.

> *"Lack of instructions for using some medications. We don't have a lot of knowledge on this and I don't know if it's something that was standardized here, for example, sometimes you prescribe a certain antibiotic without knowing if you should put it in the prescription [. . .] I don't know if it is something we should add in the prescription."* (P12)

Lack of pharmacological knowledge on drug interactions, indicated and contraindicated drugs, and dose adjustment for specific conditions were also reported. Due to lack of knowledge, one interviewee prescribed a contraindicated drug during pregnancy.

> *"Once I was prescribing for a pregnant lady, and I prescribed Fluconazole, which has proven risk to pregnancy."* (P18)

Unlike the KBMs, in the RBM the errors were not directly associated with lack of knowledge, but with lack of experience in framing the clinical situation. Through automatic thinking and using pre-acquired knowledge, the junior doctor selected routine rules which were inadequate for that specific circumstance.

> *"Sometimes there are two medications, IV and IM, and we end up quickly prescribing an IV medication as IM, out of habit. . . you rarely get an IV medication, but this is a hospital, so there are intravenous medications here where there isn't elsewhere."* (P19)

Out of habit of prescribing scopolamine associated with dipyrone to all pregnant women, one of the interviewees, unconsciously and automatically prescribed it to a pregnant patient who was allergic to dipyrone.

*"I was in a rush. . . Seeing many patients. . . You go to the pregnant patient. . ., recommend scopolamine associated with dipyrone to everyone, without knowing why and then. . . it was almost administered."* (P12)

## Violations

Violations were committed by 14 respondents and often happened when they consciously failed to comply with a rule in order to get their jobs done in time.

The computerized prescription system used at the institution makes it possible to copy the prescription from the previous day, however this prescription is removed from the system at 11 am. In order to make a copy of this prescription, the junior doctors often prescribe and release the prescription without due verification and even before evaluating the patient, which lead to several errors.

*". . . when the emergency room is full and we have to write, I don't know, thirty prescriptions until eleven in the morning. . . we end up having to copy the prescription and hand it without seeing the patient. The nursing staff keeps saying 'you have to release the prescription, the patient has no prescription. . .' and sometimes, there is no time because, if something serious comes in the emergency room, we leave to deal with the more stable patient in the infirmary when it is possible. . ."* (P12)

Many violations happened this way. Junior doctors made a copy of the prescriptions, planning to return to the system later to check and update it, but never did. The most common errors related to violations were drugs being unnecessarily prescribed and medication omission. Copying prescriptions from the previous day without properly checking them may perpetuate errors for many days.

*"The patient used AAS® and Clopidogrel® and did not receive them for 48 hours, because [the prescription] was copied and pasted, from Friday to Saturday and then again from Saturday to Sunday, and so there were no such medications in the system, and nobody noticed it. On Monday I copied and pasted it again, without these drugs, and when reviewing it, I saw the drugs were absent the previous weekend."* (P16)

Handmade alterations to printed prescriptions were mentioned in most reports. When copying and pasting a prescription into the system, changes made by hand to the printed prescription are not incorporated.

*"Antibiotic that is suspended in a clinical round, but next day the person doesn't see it was suspended by hand the day before, and ends up just copying the prescription and prescribing it that day."* (P07)

As it is part of the institution's culture to leave the preparation of medications to the Nursing team and due to the lack of knowledge on medication dilution, many respondents did not write the instructions for use of the prescribed medication, increasing the risk of errors.

*". . . an erythropoietin that had been prescribed, and then the Nursing team was 'oh god, how is it administered?', I had already left, they asked the on-duty doctor, the on-duty doctor had no idea how it was used, 'let's read the drug leaflet.' Then it wasn't administered. The next day they asked me: 'how do you use it?', so they did not do it. One day missed because of that."* (P16)

## Prescribing errors results

Out of the 105 prescribing errors reported from the perspective of the junior doctors, 42 reached the patients. Violations and slips were responsible for most of them (Fig 1). It is important to clarify that, from the junior doctors' perspective, most of the errors that reached the patients did not cause harm. These errors included omission, particularly of medicines the patients already used at home, and medicines that, for some reason, were not initiated at the appropriate time.

## Error-producing conditions

Conceptual themes have been identified from the reports and categorized according to error-producing conditions presented in Reason's Accident Causation Model [21]. The error-producing conditions are summarized in Table 2. When analyzing these conditions individually we realize that they are also interrelated.

Multiple factors were reported as error-producing conditions, but the ones which stood out in relation to the *environment* and which most accounted for slips, lapses and violations were work overload and time pressure. *Environmental conditions* such as work overload and time pressure were also the background linked to the other error-producing conditions, including the latent conditions.

**Table 2. Themes identified from the reports as error-producing conditions [21].**

| Environment | Work overload (21) |
|---|---|
| | Time pressure (16) |
| | Physical environment (4) |
| | Interruptions (2) |
| | Medical complications (1) |
| **Team** | Absent or inadequate preceptorship (9) |
| | Communication failure (4) |
| | Trust in the nursing team's supervision to prescription (3) |
| **Individual** | Lack of knowledge (13) |
| | Tiredness, sleepiness, hunger (5) |
| | Inexperience (2) |
| **Task** | Electronic prescription system (29) |
| | Handwritten prescription (13) |
| | Failure to check prescription (9) |
| | Inadequate protocol (3) |
| | Unavailable or inappropriate medical records (3) |
| **Patient** | Complex patient or patient unfamiliar to doctor (5) |
| | Polypharmacy (2) |

n = number of times the condition was mentioned by interviewees. Source: The researcher.

This table was constructed exploring the participants' speeches with the help of the Nvivo software [35].

*"Being in a hurry, many patients to see, critical patients, very long prescriptions with many items, running against time to prescribe until 11 am."* (P03)

Most junior doctors pointed out that the Emergency Room and surgical wards were unsafe environments to prescribe in.

*"[When you are] on duty, especially in the Emergency Room, where there are lots of patients to see and patients arriving, you end up not being able to check the [patient's] history since they were admitted, and not seeing that an important medication is missing, like a prophylaxis or medication that they already used at home."* (P21)

Among several reasons, the following stood out: work overload, inadequate supervision of preceptors or even lack of preceptor, which was the most cited *team-related condition*. When asked about the presence of a preceptor at the time of prescription, an interviewee replied: *"there never is one! We always do it by ourselves. . ."* (P18) The inadequate supervision by preceptors suggests devaluation of qualified training in service, which is the main objective of the residency.

Poor communication, especially between the medical and nursing teams, was cited in the reports. Despite this, many interviewees trusted that the nursing staff would detect errors, and disregarded aspects of the prescription such as checking and instructions for use of the medications. The nursing team has an important role in detecting the prescription error, however, as they do not participate in the planning of the prescription, they would not detect, for example, a mistake. Another factor is that many nurses are also trainees in the nursing undergraduate course of the institution and are inexperienced.

*"We're used to it because the nursing team is very well trained, they know how to dilute and all. So sometimes this error happens because of that." (P08)*

*Environmental conditions* and *team* naturally relate to each other. Work overload and time pressure may have contributed for the preceptors, who were also out of time due to work overload, to not pay proper attention to the junior doctors, leading them to commit unsafe acts. In this sense, we can think that unintentional violations committed by junior doctors may have been "copied" from the behavior of their preceptors' when dealing with similar situations. The same may have occurred with the nursing team that, overloaded with work and without time, did not communicate properly with the medical team.

Unsafe acts such as RBMs may have been influenced by the *environment* and *team*, as the routine was mistakenly considered the norm, which forced both junior doctors and their preceptors to behave automatically in situations that should have been handled individually with more time and thought.

Lack of knowledge and inexperience were reported as the factors most accountable for mistakes. Inadequate preceptorship further exacerbates this condition. Lack of knowledge about medication dilution was the most cited mistake.

*"The most common mistake I made during my training was getting the drug dilution wrong, especially when the patient is sedated or uses vasoactive drugs. Sometimes we get the dilution wrong!" How many ampoules should you dilute it in, how much the drip should run and for how long." This has already happened a couple of times."* (P18)

*Individual conditions* such as mistakes caused by lack of knowledge and inexperience are also related to *environmental* and *team conditions*. Work overload and time pressure may have led junior doctors to risk prescribing without clarifying their concerns, due to the lack of a preceptor. We can assume that this situation may also be a violation, since the junior doctors were aware of the risk of making mistakes.

The most prominent *task-related condition* was the electronic prescription system. Prescribing by hand and not checking whether the prescription was in line with current clinical assessments also stood out. Absence of protocol and conflict between conducts were also mentioned.

> *"Protocols are being established now, but if a boss is from one segment and the other from another, or if the shift changes, all practices change. I'm having a lot of difficulty with this at GO [Gynecology and Obstetrics]."* (P14)

The *task-related condition* of not checking whether the prescription was in line with current clinical assessments was considered a violation, possibly caused by *environment conditions* such as work overload and time pressure. Absence of protocols and task conflicts is also related to the *team-related condition*. Other factors linked to *patients*, such as unknown and unstable patients and patients with severe cases who used multiple medications were also reported.

## Latent conditions

Emergent themes that have been identified from the reports related to latent conditions are summarized in Table 3. According to Reason [21], latent conditions are relevant because "unlike active failures, whose specific forms are often hard to foresee, latent conditions can be identified and remedied before an adverse event occurs" and "we cannot change the human condition, but we can change the conditions under which humans work". The latent conditions relate to each other and to the error-producing conditions analyzed earlier.

Organizational conditions related to the prescription process and the flawed electronic system were latent conditions that stood out. These conditions, associated with the perceived low importance given to prescriptions, in particular to prescription copies and low awareness about the errors perceived during the interviews (difficulty to remember the errors) further increase the risks of unsafe occurrences.

Another latent condition which stood out was a culture of exploitation of the junior doctor's workforce, suggested in many reports of work overload. Work overload and gaps in the health system are directly related. Some interviewees express this cultural issue in relation to the use of junior doctors' work to fill gaps in the health system. Besides being used as a workforce, they are still subjected to inadequate working/training conditions, such as the lack of an

**Table 3. Themes identified from the reports as latent conditions [21].**

| |
|---|
| Organizational problems, a flawed electronic system and the prescription process |
| Exploitation of the junior doctor's workforce |
| Low importance given to execution of prescription form |
| Priority to practice training especially in clinical surgery programs |
| Insufficient learning (absence) on prescribing medications during medical residency course |
| Insufficient learning (absence) on prescribing medications during the undergraduate course |
| Learning difficulties related to fear of devaluation by the supervisors/medical hierarchy |

This table was constructed exploring the participants' speeches with the help of the Nvivo software [35].

appropriate place to serve patients. In fact, the institution where the study was carried out is a teaching hospital and a service provider within the Brazilian Unified Public Health System, which despite being a good example of a health system with principles such as universal, equitable, and integral care, is going through a difficult economic period with budget cuts and an increased demand for care. It is common, particularly in the emergency room, for patients to be placed on stretchers and chairs in the corridors.

> *"We, as junior doctors, having more preceptors, having to be less in charge of the work and have more discussions, I think that would be the way." (P06)*

> *". . . There is no office! The patient is seen in his seat, if you are lucky to have a stretcher, they are seen on the stretcher, if not, the exam is in the seat, there is no way to touch the abdomen, you ask embarrassing questions in front of everyone in the room . . . This is very serious, we have no structure to prescribe, nor to diagnose, nor anything." (P16)*

The low importance given to the execution of the prescription form was perceived by the authors in many reports of failure to check the prescription and in the omission of instructions of use in the prescriptions, even when they were aware of the risks. This condition, associated with work overload, time pressure and inadequate preceptorship, further aggravates the possibility of the occurrence of unsafe acts. This low importance is certainly related to the condition of insufficient learning on prescribing medication. The interviewees may have deduced that if it was not taught, it is not so important. We believe that some interviewees only became aware of the importance of a safe prescription when, through the interviewer's questioning and feedback, they were stimulated to reflect.

Priority for practical training in surgical specialties, and the short time of residency also seemed to be a latent cause for some respondents to prioritize other areas of learning and consider the prescription of medications less important. The preceptors themselves may also be unconsciously passing on this erroneous information, as they are present during surgeries but according to most reports, are not present at the time of prescription.

> *". . . the surgical environment is stressful, the intraoperative, the preoperative. Residency in general surgery is one of the [specialties] that demand the most from junior doctors. With a [high] workload, sometimes out of choice, [because] we know that two years is a short time,—orthopedics and other surgical specialties depend on you doing it, books don't teach you how to operate, you learn by doing -, sometimes even if you are the most dedicated [junior doctor], you miss something."* (P19)

Insufficient learning on prescribing medications during the undergraduate course and inadequate training during residency were reported by most junior doctors as latent conditions for the occurrence of most mistakes. Considering that they graduated from 12 different medical schools in six different Brazilian states, we might think that this is the Brazilian reality as regards undergraduate programs.

> *"I think we should have a course just to learn how to prescribe, because at least here at the internship, it's like this: the intern doesn't write it, someone dictates it to the intern without us knowing what it's for, how it works, how to prescribe it. Is this the dose used, is it just for this patient? So, if you want to learn, it's a matter of chasing them and keeping asking."* (P10)

Medical hierarchy and the inappropriate behavior of some preceptors made junior doctors feel embarrassed to seek help when they were unsure, which may have contributed to more

unsafe acts. This condition contributes to the fact that they avoid consulting their preceptors and prescribe unsafely, running the risk of committing planning failure, such as KBM and RBM. On the other hand, we can say that it is also a violation of the rule since they were aware of the risk of making mistakes due to lack of knowledge.

> *"You'll get there and see the two bosses sitting. They don't get up to look at the patient's face, and are extremely grudging when you ask for help to prescribe something."* (P16)

## Discussion

The results showed similarities with international studies [4, 17, 18, 22, 24–27]. Most reported errors were caused by slips and lapses, followed by mistakes. However, unlike in the international studies, there was a large number of errors resulting from violations, and these violations were responsible for most errors that reached the patients, without harm, according to the reports.

Work overload, organizational and managerial problems of the institution (such as inadequate prescription systems and training), combined with the country's cultural, socioeconomic and political reality (health system and public policies) and misuse of the junior doctors' work to fill gaps in the system may have accentuated this picture.

The violations detected in the study were not seen and mentioned by the interviewees as violations, but as difficulties in their routines, tasks and prescription protocols, i.e., difficulties in performing the prescribed work, which led the prescribers, as explained by Dejours [41], to "cheating" the norm in order to be able to perform the task. According to the author, for a better understanding of human errors in executing tasks, the concepts of "prescribed work" and "real work" are relevant. Real work is defined as "what the worker must add to the prescription to achieve the objectives assigned; or even what they must add of themselves to go round what does not work even if the rules are followed" [41]. Breaking rules intentionally is a cause for concern, as it does not occur without risks, particularly in the context of health organizations, where there are risks of making mistakes and causing harm to patients.

The most common violation in the reports was intentionally failing to follow a rule in order to carry out the prescription on time, suggesting that work overload and time pressure played a great role in such occurrences. According to Runciman, Merry & Walton [42], this type of violation can be explained as "shortcuts to perform a routine task", i.e., "routine violations" and, despite being deliberate, one may agree with Reason [43] that they are not necessarily reprehensible, since work overload was the main condition for their occurrence. However, Runciman, Merry & Walton [42] suggest that unsafe acts of routine violation are only justified if they are not frequent, which raises the question of whether, in this situation, the junior doctor should be reprimanded.

One way to try to resolve this matter may be through the "Just Culture" model [44]. It attempts to differentiate, particularly in the context of rule violations, the results from acts of negligence or recklessness, from the results of an unsafe act that could be worked on. Reckless behaviors, such as acting in disregard of rules and being aware of the risks would be the only ones liable to blame and reprimand [45].

On the other hand, Reis-Dennis [46] argues that the "Just Culture" model presents problems resulting from a limited understanding of the value of punitive practices used to hold people accountable for their actions. In his study, the author demonstrates that the type of "accountability" and "punishment" advocated by the "Just Culture" disrespects both patients and health professionals.

Some junior doctors believed their errors would be detected and stopped by other health care professionals, which may have accentuated the practice of not checking prescriptions (violation). Failure to check prescriptions and prescriptions made by hand were the most cited task-related condition, which suggests that the studied hospital does not have a safety culture in relation to prescriptions.

The most frequently mentioned types of prescribing errors were drug omission and unnecessary prescription. Omissions are among the most relevant types of drug incidents in the world due to their high frequency and potential harm to the patient [47, 48].

In our study, the lack of and inadequate preceptorship was widely reported in pressure-laden environments. Individual supervision from experienced doctors help to reflect and recognize where they may have made mistakes, particularly when on duty and in different training locations [49].

Difficulty to remember or even to identify errors at the beginning of the interview points to the possibility of little awareness about the errors themselves, which could be improved with feedback or team reflection.

Considering the several accounts of work overload and of stress for dealing with multiple tasks, it can be said that the results of the studies by Sponholz et al. [50] apply to our hospital and probably to other public teaching hospitals in Brazil and other developing countries whose public health systems operate with limited resources. Such studies reveal long weekly working hours, continuous hours of work, lack of supervision and high stress load among junior doctors. The authors concluded that the work process is characterized by the subordination of teaching and learning to the exploitation of the junior doctors' workforce. These analyses indicate deeper social and institutional causes. Further studies should be carried out in order to clarify this hypothesis.

In the present study, lack of knowledge was the most discussed factor causing mistakes, and inadequate and/or absent preceptorship may have aggravated the situation. Mistakes are harder to be discovered, especially by other non-medical health professionals, because the objective and the intended path for pharmacological treatment are often unknown, hence the importance of checking the junior doctor's prescription by the preceptor in charge.

In most cases, knowledge-based mistakes involved lack of knowledge regarding the dose, dilution, route of administration and drug interaction, all practical aspects of the prescription which junior doctors might not necessarily know, but which could be corrected through seeking information from other sources, such as pharmacists.

Rule-based mistakes generally involved inexperience in situations involving allergic patients, unnecessarily prescribed medications and dose adjustment according to the patient's weight. In these situations, the junior doctors generally prescribed the medication and/or dose used in "standard" conditions, thinking they were correct, but were inadequate for the circumstance.

Regarding the occurrence of mistakes, it should be noted that the majority of participants did not receive prescription training during their undergraduate courses nor were they evaluated on this skill, which helps to explain the result. Some international medical universities have developed a unique pharmacotherapeutic test to assess whether future prescribers can prescribe safely [51].

It is also important to consider that the change in prescription behavior will only occur if junior doctors are aware of their errors [52]. Junior doctors, experienced doctors and pharmacists are interdependent, and their interaction provides a means for developing effective prescribing skills. This interdependence and feedback are both an opportunity to improve the education and training of junior doctors and update doctors in general [53].

Considering that, in the present study, the electronic prescribing system was the most highlighted error-producing and latent condition, it is suggested that the design of the electronic prescribing system used may be contributing significantly to the occurrence of errors. Improvements in this system should be considered a priority.

In our study, apart from the large number of reported errors reaching patients, many other errors were detected in reports from nurses and pharmacists, signaling that interventions to improve such barriers would also be valuable to prevent errors from reaching patients. In this sense, in addition to implementing measures to avoid errors, interventions to safely intercept them are also beneficial [16].

Regardless of a country's level of wealth, healthcare professionals must offer care with greater safety. Principles of patient safety should be applied, regardless of the professionals involved, the place where care is provided and the type of patient needing care [54]. Although increasing the number of professionals and financial resources is a necessary measure, it is not sufficient to reduce harm to patients [55]. Despite the social implications related to the deficient health system evidenced in this study, it is possible to elaborate good strategies to reduce prescription errors.

## Strengths and limitations

As memory may be deficient, the interviewees' memories might have been affected by present events [56]. Attribution bias to escape accountability for the error may have occurred and led the participant to attribute their errors to external factors rather than to themselves. However, the types of errors reported and their possible causes are comparable with other studies [17, 19, 24–27].

Despite these limitations, methodologies proposed in international studies were used, enabling a comparison in relation to the understanding of the complexity of prescribing errors. Also, although the study was carried out in a single location, the facts that there were doctors from six different universities in Brazil and a variety of prescribing environments added credibility to the results.

The methodology used allowed the identification of events by the participants themselves. In the case of prescribing errors, the junior doctors themselves could correct errors which might not be identified by other professionals. Another advantage and differential of the study was that it considered the junior doctors' perspective on errors made and/or identified in prescriptions by other junior doctors. Through these reports, it was possible to discuss prevalent errors, and approach situations which the junior doctors might not have reported due to embarrassment.

## Conclusions

Even though the results of this research are based on the interviewees' perspectives, they reaffirm international findings about the severity and nature of prescribing errors in hospital settings and the multifactorial aspect of their causes. Unlike other studies, however, we found a large number of violations. The active failures and their producing and latent conditions tend to be related, on the interviewees' perspectives, to the socioeconomic, cultural and political reality of the studied site and to the profile of the problems detected.

Considering the third WHO's global patient safety challenge knowing these patterns is important for implementing strategies to control such failures.

The results also suggest the implementation of interventions to improve the safety of both patients and professionals involved in the medication process, in particular a reassessment and monitoring of the computerized prescription system.

As changes in the national health system are not achievable in the short term, it is suggested that the culture of exploitation of the junior doctors' work be discouraged, that their training be prioritized, and that a culture of safety in drug prescribing be encouraged. Other relevant suggestions are: (a) Raising awareness among hospital managers and coordinators of the medical residency program about the priority of training and reducing work overload and also about the importance of training and of the effective and responsible participation of preceptors in the training; (b) Improving defense barriers in relation to errors with the inclusion of pharmacists in clinical visits, providing 24-hour access to a pharmacist and implementing specific prescription protocols; (c) Integrating the patient safety discipline into the undergraduate and residency curriculum; (d) Improving the theoretical and practical teaching on prescription in undergraduate courses, and considering the assessment of this skill as a requirement for exercising the profession; (e) Readapting the discipline of pharmacology to something close to the reality of prescription drugs; (f) Valuing multidisciplinary work with regard to the prescription of medicines with feedback from preceptors and pharmacists; (g) Prioritizing risk management actions, routine monitoring and reevaluation of the prescription system and institutional prescription protocols and, finally, implementing measures that value the importance of prescription and notification of errors as a safety measure for patients and doctors in general.

Other local or multicenter studies investigating contextual cultural aspects that help to understand the causes of prescription errors in Brazil and in other middle and low income countries need to be carried out.

## Supporting information

**S1 File. Interview topic guide in Portuguese.**
(ZIP)

**S2 File. Interview topic guide in English.**
(ZIP)

## Acknowledgments

The authors thank all the junior doctors who offered their time to take part in this study.

## Author Contributions

**Conceptualization:** Gislaine F. Bonella, Alicia R. N. D. de Souza, Carlos H. M. da Silva.

**Data curation:** Gislaine F. Bonella, Letícia da S. Alves.

**Formal analysis:** Gislaine F. Bonella, Letícia da S. Alves, Alicia R. N. D. de Souza, Carlos H. M. da Silva.

**Investigation:** Gislaine F. Bonella, Letícia da S. Alves.

**Methodology:** Gislaine F. Bonella, Letícia da S. Alves, Alicia R. N. D. de Souza.

**Project administration:** Gislaine F. Bonella, Carlos H. M. da Silva.

**Software:** Gislaine F. Bonella.

**Supervision:** Alicia R. N. D. de Souza, Carlos H. M. da Silva.

**Validation:** Alicia R. N. D. de Souza, Carlos H. M. da Silva.

**Writing – review & editing:** Gislaine F. Bonella, Letícia da S. Alves.

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
