## [Decision Letter · Decision Letter 0]

27 Apr 2021

PONE-D-21-03374

Prescribing errors in a Brazilian teaching hospital: causes and underlying factors from the perspective of junior doctors

PLOS ONE

Dear Dr. Bonella,

Thank you for submitting your manuscript to PLOS ONE. After careful consideration, we feel that it has merit but does not fully meet PLOS ONE’s publication criteria as it currently stands. Therefore, we invite you to submit a revised version of the manuscript that addresses the points raised during the review process.

We look forward to receiving your revised manuscript.

Kind regards,

Khatijah Lim Abdullah, DClinP, MSc., BSc

Academic Editor

PLOS ONE

Journal Requirements:

2. Please provide additional details regarding participant consent. In the ethics statement in the Methods and online submission information, please ensure that you have specified what type you obtained (for instance, written or verbal, and if verbal, how it was documented and witnessed).

In addition, in the ethics statement in the manuscript Methods, please specify whether participation in the study was voluntary, and whether participants could opt out at any time. Please also state whether participation was encouraged by any particular incentives, and any steps that were taken to ensure that participants did not feel pressurized to participate. Finally, please state whether the ethics committee approved all the recruitment and consent procedures.

3. Please include a copy of the interview guide used in the study, in both the original language and English, as Supporting Information, or include a citation if it has been published previously.

Additional Editor Comments :

Dear Authors

Please include a section on how the rigor of the data analysis was ensured , and each quotes should have an anonymized details

Reviewers' comments:

Reviewer's Responses to Questions

**Comments to the Author**

1. Is the manuscript technically sound, and do the data support the conclusions?

Reviewer #1: Yes

Reviewer #2: Partly

2. Has the statistical analysis been performed appropriately and rigorously? 

Reviewer #1: N/A

Reviewer #2: N/A

3. Have the authors made all data underlying the findings in their manuscript fully available?

Reviewer #1: Yes

Reviewer #2: Yes

4. Is the manuscript presented in an intelligible fashion and written in standard English?

Reviewer #1: Yes

Reviewer #2: Yes

5. Review Comments to the Author

Reviewer #1: Originality: Analysis of prescribing errors caused by junior doctors have not been adequately evaluated in Brazil, therefore this study is original.

Significance: Determining the burden and characteristics of prescribing errors is fundamental to designing appropriate measures in order to mitigate the risks and harms to the patients, especially a large number of violation errors were detected in this study. I agree that pharmacists’ perceptions during the support in clinical decisions when prescribing can increase the safety of medication uses.

Methodology: There are some suggestions in methodology:

(1). This study was conducted in one teaching hospital as compared with seven teaching hospitals as performed by Lewis PJ and Ashcroft DM et al. (2014) in UK, more similar study sites should be recruited to increase the generalizability.

(2). Selection bias was likely to occur due to the no responses from the 632 junior doctors. The low response rate for the recruitment can be reduced when more study sites are employed. Also, surface mail for the invitation may be the other solution.

(3). Independent assessors, both authors G and L were involved in the semi-structured interview that selection bias may occur. Instead, well-trained independent interviewers should be employed. Also, third external independent assessor should be used if there are discrepancies between authors G and L regarding the classifications of different types of prescription errors, unsafe acts and the latent and error-producing conditions.

Accuracy and quality of results: Well organized table and figures were used for the data presentation. Annotation of the abbreviations of KBM and RBM should be used for Figure 1.

Clarity and lucidity of presentation: The authors were able to provide plausible explanation for the results of this study. Since electronic prescription system constituted a remarkable number (n=29) of the error-producing conditions reported by junior doctors, enhancement of the prescription and electronic medical records can be mentioned at the discussion part.

Adequacy of references: The references included are adequate. However some incorrect referencing numbers are found.

Reviewer #2: Thank you for inviting me to review this manuscript which describes the prescribing errors in a brazilian teaching hospital.

Overall, I find the manuscript somewhat confusing. The authors have used a qualitative approach to explore the causes of medication prescribing errors among junior doctors. However, the results presented are very quantitative. Table 1 and 2. The manuscript is also very wordy. The authors should consider cutting down on the number of words, and make their writing more concise.

Introduction

I like the model (i.e. Figure 1) which is used to explain the conceptual framework of this study. However, the authors tend to repeat what is in the figure as text. Please avoid this. Conceptual framework should be summarised, and jusitification for selecting this theory should be provided The authors should provide a stronger justification on why this work is needed. Note, that any abbreviations used in the figure (e.g. KBM) should be explained as a footnote. Aim should be changed to explore .... (rather than investigate - which is very quantitative)

Methods

Should use more subheadings in this section to aid reading. How was the topic guide developed? should include the questions asked as a table. Where's the rigour and reflexivity sections?

Results

Lines 235-242 should be presented as a table to decrease the number of words.

Table 2: how did the authors collect this data? Did they use a survey form? Very quantitative results

For a qualitative paper, I do not see what themes emerged from the participants. I only see a repetition of the conceptual framework (i.e. results are presented according to the conceptual framework). As such, I don't think the authors actually explored why the junior doctors made the prescribing errors. Also, the quotes do not have the annonymised details of the junior dr. My general impression is that the authors have not analysed their data in depth sufficiently for a qualitative study.

Discussion

Too wordy. Needs to be summarised.

6. PLOS authors have the option to publish the peer review history of their article (what does this mean?). If published, this will include your full peer review and any attached files.

Reviewer #1: **Yes: **CHUI Chun Ming William

Reviewer #2: No

---

## [Author Response · Author response to Decision Letter 0]

10 Jun 2021

Response to Reviewers

Dear Editor and Reviewers,

We appreciate the suggestions.

We believe we’ve addressed all of them and that it has greatly improved our manuscript. 

Thank you!

Journal Requirements

Response: Yes, we have reviewed our manuscript and believe it conforms to the PLOS ONE templates. 

If any details are still in disagreement please allow us to correct them.

2: Please provide additional details regarding participant consent. In the ethics statement in the Methods and online submission information, please ensure that you have specified what type you obtained (for instance, written or verbal, and if verbal, how it was documented and witnessed).

Response: We agree with the suggestions and added: “All participants gave their written informed consent before participating. Ethical risks were minimized by adopting total confidentiality regarding the data and identity of the participants (anonymity through impersonal coding), and through rigorous care when explaining and obtaining participants’ consent”. (lines 175 to 178)

 In addition, in the ethics statement in the manuscript Methods, please specify whether participation in the study was voluntary, and whether participants could opt out at any time. Please also state whether participation was encouraged by any particular incentives, and any steps that were taken to ensure that participants did not feel pressurized to participate. Finally, please state whether the ethics committee approved all the recruitment and consent procedures. 

Response: We agree with the suggestions and added: “The study participant was informed through the written consent form and the interviewer that their participation was voluntary, had no financial expense or gain and that they were free to opt out at any time, without any harm or coercion. The interviews were conducted in a private place, out of working hours and the audio was recorded. After being transcribed the audio files were deleted (lines 178 a 182)

“The study, recruitment and consent were approved without restrictions by the Human Research Ethics Committee of the Federal University of Uberlandia on 03/28/2018, under process number 2.570.103.

All of this information was also in the consent form. See below the model used.

FREE AND INFORMED CONSENT FORM

You are being invited to participate in the research titled " Prescribing errors from the perspective of junior doctors", under the supervision of researchers Prof. Dr. Carlos Henrique Martins da Silva and Gislaine Ferraresi Bonella, who is also a pharmacist at HC-UFU. 

In this research we are seeking the perspective (views, beliefs, and experience) of physicians on the possible causes and potential consequences of prescribing errors.

The Free and Informed Consent Form will be collected by the researcher Gislaine. In participating, you will be interviewed about your opinions, recollections, and experiences related to prescribing errors.

With your permission, the interview will be audio recorded. You can request that the recorder be turned off at any point during the interview, which will last between 30 minutes. The interview and the collected data are private and will be kept completely confidential. The audio will be erased after transcription and you will not be identified at any time. Even when publishing the results of the research, your identity will be kept private. The risks refer to confidentiality, secrecy, and privacy. To avoid them, the executing team is committed to complete confidentiality of the data and the identity of the participants (for which we will use anonymity), and to the erasure of the interview after transcription. No information will be used professionally regarding the participants.

The immediate benefits will be knowledge, reflection and better understanding of the subject. Other benefits are the development/improvement of organisational protocols and systems to prevent prescribing errors in order to ensure greater physician and patient safety.

You will have no expenses and no financial gain by participating in the research and you are free to opt out at any time without any loss or duress.

You will keep an original copy of this Informed Consent Form.

Any questions regarding the research, you may contact: Prof. Dr. Carlos Henrique Martins da Silva, at the Office of the Medical School, Campus Umuarama, Av. Pará, nº 1720, Uberlândia - MG, CEP: 38405-320; phone: 34-3225-8625. You can also contact CEP - Committee for Ethics in Research with Human Beings at the Federal University of Uberlândia: Av. João Naves de Ávila, nº 2121, bloco A, sala 224, Campus Santa Mônica - Uberlândia -MG, CEP: 38408-100; phone: 34-3239-4131. The CEP is an independent collegiate created to protect the interests of research participants in terms of their integrity and dignity and to contribute to the development of research within ethical standards in accordance with the resolutions of the National Health Council.

Uberlândia, ....... of ……..of 201.......

Researchers' signatures

I agree to voluntarily participate in the above mentioned project after having been duly informed.

Research Participant

3. Please include a copy of the interview guide used in the study, in both the original language and English, as Supporting Information, or include a citation if it has been published previously. 

Response: We agree with the suggestions and added as supporting information “S1 and S2”.

Additional Editor Comments : 

Dear Authors Please include a section on how the rigor of the data analysis was ensured, and each quotes should have an anonymized details

Response: We agree with the suggestions and included a section “Rigor of data analysis” (lines 219) and the quotes were referenced using numbers representing the participant (P) code, eg, P 01, P02, etc. (lines 254 to 255)

Reviewers' comments:

Reviewer #1: Originality: Analysis of prescribing errors caused by junior doctors have not been adequately evaluated in Brazil, therefore this study is original. 

Significance: Determining the burden and characteristics of prescribing errors is fundamental to designing appropriate measures in order to mitigate the risks and harms to the patients, especially a large number of violation errors were detected in this study. I agree that pharmacists’ perceptions during the support in clinical decisions when prescribing can increase the safety of medication uses. 

Methodology: There are some suggestions in methodology: (1). This study was conducted in one teaching hospital as compared with seven teaching hospitals as performed by Lewis PJ and Ashcroft DM et al. (2014) in UK, more similar study sites should be recruited to increase the generalizability.

 (2). Selection bias was likely to occur due to the no responses from the 632 junior doctors. The low response rate for the recruitment can be reduced when more study sites are employed. Also, surface mail for the invitation may be the other solution.

Response: We agreed with the suggestion. As we did not find similar studies in Brazil we decided to start the research primarily in a university hospital in order to familiarize ourselves with this type of study and then expand the study to include other university hospitals throughout Brazil to increase generalizability.

 (3). Independent assessors, both authors G and L were involved in the semi-structured interview that selection bias may occur. Instead, well-trained independent interviewers should be employed. Also, third external independent assessor should be used if there are discrepancies between authors G and L regarding the classifications of different types of prescription errors, unsafe acts and the latent and error-producing conditions.

Response: We agreed with the suggestions and for further studies we intend to conduct training for the interviewers and add a third external independent assessor.

Accuracy and quality of results: Well organized table and figures were used for the data presentation. Annotation of the abbreviations of KBM and RBM should be used for Figure 1.

Response: We have included the annotation of the abbreviations in figure 1.

Clarity and lucidity of presentation: The authors were able to provide plausible explanation for the results of this study. Since electronic prescription system constituted a remarkable number (n=29) of the error-producing conditions reported by junior doctors, enhancement of the prescription and electronic medical records can be mentioned at the discussion part.

Response: We agree with the suggestion and added to the discussion a reflection on the need for enhancement of the prescription and electronic medical records (lines 618 to 621). “Considering that, in the present study, the electronic prescribing system was the most highlighted error-producing and latent condition, it is suggested that the design of the electronic prescribing system used may be contributing significantly to the occurrence of errors. Improvements in this system should be considered a priority”.

Adequacy of references: The references included are adequate. However, some incorrect referencing numbers are found.

Response: We have agreed and corrected the references. 

Reviewer #2: Thank you for inviting me to review this manuscript which describes the prescribing errors in a braziian teaching hospital. 

Overall, I find the manuscript somewhat confusing. The authors have used a qualitative approach to explore the causes of medication prescribing errors among junior doctors. However, the results presented are very quantitative. Table 1 and 2. The manuscript is also very wordy. The authors should consider cutting down on the number of words, and make their writing more concise.

Response: We agree with the suggestions and have improved the text to avoid confusion. We agreed that it is not appropriate in qualitative research to identify percentages and it was not our aim in the study to emphasize such aspects, so we removed them from the text and tables. We also reduced the number of words in the manuscript. 

Introduction I like the model (i.e. Figure 1) which is used to explain the conceptual framework of this study. However, the authors tend to repeat what is in the figure as text. Please avoid this. Conceptual framework should be summarised, and jusitification for selecting this theory should be provided The authors should provide a stronger justification on why this work is needed. Note, that any abbreviations used in the figure (e.g. KBM) should be explained as a footnote. Aim should be changed to explore .... (rather than investigate - which is very quantitative)

Response: We agree with the suggestions and removed the repetition from the text. We have summarized the conceptual framework (table 1) and justified why we use “Reason’s Accident Causation model” in our study: We “have used “Reason’s Accident Causation model” as it addresses behavioral factors linked to errors, enables integrating individual and systemic approaches, and is widely used in hospital contexts [3,4,17,18, 22-29]”(lines 79 to 81).“The problem of prescription errors seen in an individual manner is associated, in part, with human behavior and its failures. This, in turn, is related to a cognitive and emotional structure not easily understood by quantitative research alone. Hence, we chose a qualitative approach to investigate subjective aspects related to errors and an analysis framework based on Reason’s Accident Causation model” (which approaches an individual and systemic view of errors) (lines 97 to 101). In addition, we have provided a stronger justification for why this study is needed: “In Brazil no qualitative study carrying out an in-depth investigation of the underlying causes of prescribing errors from the perspective of junior doctors has been found. This research is made necessary as it will be useful for clinicians, mainly physicians, nurses, and pharmacists, whose professions are directly linked to medication prescription errors, and also contribute to reducing the risk of harm to patients. Although other international studies have addressed this perspective, the same type of investigation in a different and not yet properly explored environment will, on one hand, contribute to a better global understanding of the problem and, on the other, will connect to problems and the specific socio-cultural reality of developing countries, where interventions are necessary and sought after” (lines 105 to 114). We changed the word investigate to explore.

Methods Should use more subheadings in this section to aid reading. How was the topic guide developed? should include the questions asked as a table. 

Where's the rigour and reflexivity sections?

Response: We agree with the suggestions and have added more subheadings in the methods section, including a section for:

Qualitative approach and research paradigm 

We conducted a qualitative, descriptive exploratory study, collecting data through individual interviews. Focusing on the subjective and personal dimension of human error, we adopted a constructivist paradigm, assuming that there are multiple and distinct subjective realities (perspectives) [31]. Within this paradigm, we adopted for concept-based exploration of themes, Reason’s Accident Causation model [21]. The epistemic dimension was phenomenological [31], starting from the way reality appears to the participants themselves, taking them as autonomous subjects of medical practice. (lines 119 to 127)

Researcher characteristics and reflexivity

The author who conducted all the interviews was not known or acquainted with the participants before starting the study. The medical prescriptions are usually printed directly in the pharmacy sector and when a problem is detected, the pharmaceutical team contacts the doctor by phone for clarification. No member of the study taught, supervised, or had any authority over participants in the study. 

The interviewer was fully aware of her potential assumptions during the interviews, taking care that these did not interfere with the accounts, she tried not to deviate from the questions contained in the interview guide and also tried to ensure that all data could be reviewed and discussed by the entire research team and that disagreements could be discussed and resolved by consensus ensuring neutrality and objectivity to the study. (lines 129 to 140)

Context 

The study was carried out in a teaching hospital and service provider of the Brazilian Unified Public Health System (Sistema Único de Saúde - SUS) containing 530 hospital beds and served by a computerized prescription system. This system does not allow the locking of the prescribed medications, but it does have a maximum dose alert for some types of medications, such as high-risk medications. 

In our study, the term junior doctor refers to the doctors who, after finishing their undergraduate degree, go on a residency program to specialize in an area of medicine. The vast majority are newly graduated doctors. (lines 142 to 151)

Sampling strategy 

The sampling universe consisted of doctors from the residency program who were training at the hospital in the period between April and July 2018. First, second and third year residents were included in the study, those with over three years and those on vacation or on sick leave during the study period were excluded. 

Invitation to participate in the study was initially done through an email to all (632) junior doctors enrolled in the Medical Residency Program, however there were no responses. Thus 40 participants were individually recruited in their workplace, either in person or by telephone. As one of the aims of the study was to identify contextual causes and previous errors in different hospital environments, junior doctors of both genders and from different settings (wards, adult and pediatric intensive care units and emergency care units) were selected. The study also sought to contemplate several medical specialties. None of the recruited participants refused to participate in the study, however, six did not participate due to their impossibility to schedule appointments with the interviewer, which, according to them, was due to their residency schedule and work overload. Theoretical saturation was obtained with 34 interviews [32]. (lines 153 to 169)

Ethical issues

The study, recruitment and consent were approved without restrictions by the Human Research Ethics Committee of the Federal University of Uberlandia on 03/28/2018, under process number 2.570.103. All participants gave their written informed consent before participating. Ethical risks were minimized by adopting total confidentiality regarding the data and identity of the participants (anonymity through impersonal coding), and through rigorous care when explaining and obtaining participants’ consent. The study participant was informed through the written consent form and the interviewer that their participation was voluntary, had no financial expense or gain and that they were free to opt out at any time, without any harm or coercion. The interviews were conducted in a private place, out of working hours and the audio was recorded. After being transcribed the audio files were deleted. (lines 171 to 182)

Data collection, instrument and processing 

An interview topic guide was used with questions related to the “planning” and “execution” of the prescription (S1 and S2), and was an adaptation of the one used in Lewis et al. [26], with due authorization. This approach provided a detailed account of prescribing decisions (specific behaviors when planning and executing the prescription) [26].

The interviews were conducted by the first pharmaceutical author G, as she has over 20 years of experience in hospital pharmacy, routinely dealing with prescription errors, at a time chosen by the participant and respecting his or her availability. The interviews lasted between 15 and 35 minutes and were audio-recorded and transcribed in full.

In the interview the participant was asked to report prescribing errors he/she had committed and/or errors detected in prescriptions by other junior doctors. The definition of error used and shared with participants was “when, as a result of a prescribing decision or prescription writing process, there is an unintentional, significant reduction in the probability of treatment being timely and effective or increase in the risk of harm when compared with generally accepted practice” [33]. (lines 184 to 199)

Data analysis 

For a qualitative investigation exploring subjective perspectives we used a Framework Approach [34] for data analysis. A period of familiarization with the raw data (listening to the recordings, reading the transcripts, checking notes with reflective care) was previously undertaken by G and L to ensure that they had relevant information for the study's purpose. Reason’s Accident Causation model [21] was used to categorize and present the data. The development of the analytic framework and a categorization and coding according to “Reason's Accident Causation model” were carefully elaborated by G. This model was the most commonly used theoretical model when considering prescribing errors [3,4,17,18,22-29]. To ensure higher accuracy, the application of this framework and the mapping of the data were performed independently by G and L and the differences were settled by consensus. Tables and organizational charts were constructed to facilitate understanding and interpretation. 

The computer software program NVivo© [35] was used to assist in the organization of the data. Codes and sub-codes to qualify the 34 cases were previously defined and used during the analysis process to identify potential themes. Prescribing errors were then counted and named according to the taxonomy in Otero-López et al. [36]. (lines 201 to 217)

Rigor of data analysis

We sought to link the data analysis to theoretical constructs, theories used in preliminary work, and codes derived from the theoretical framework of relevant literature to ensure greater reliability and credibility. National and international recommendations/protocols on good prescription practice and patient safety [37 – 40] were used as objective aspects to guide our interpretation and critical discussion of the data to avoid evaluative bias. To ensure researcher neutrality and objectivity, all data was made available for approval by all the authors, two of whom were senior doctors and experienced researchers in qualitative research (A) and health sciences (C). Once the findings were sufficiently clarified with the chosen structural analysis, the interaction between findings and theories enabled transferability. The relevant findings and interpretations present applicability beyond the greater understanding of the problem, such as encouraging new research (in Brazil and in other countries), promoting critical-reflective evaluation and suggesting improvements in care, benefits for the participants and improvements in patient safety. (lines 219 to 233)

How was the topic guide developed? should include the questions asked as a table. 

The questions are in the topic guide adapted from the model used in the Lewis et al. [26] study and are now attached as additional files (S1 and S2), so we believe there is no need to include the questions asked as a table in order not to overload the manuscript.

Results Lines 235-242 should be presented as a table to decrease the number of words.

Response: We accepted the suggestion and created a table (Table 1) 

Table 2: how did the authors collect this data? Did they use a survey form? Very quantitative results For a qualitative paper, I do not see what themes emerged from the participants. I only see a repetition of the conceptual framework (i.e. results are presented according to the conceptual framework). As such, I don't think the authors actually explored why the junior doctors made the prescribing errors. 

Response: Regarding Table 2 (currently it is Table 4), we collected the data through semi-structured questions included in the topic guide (S1 and S2) that referred to this aspect, identifying words and phrases such as: 

Related to the environment: “Being in a hurry” (time pressure), “running against time” (time pressure), “amount of work” (work overload), “amount of patients”, “lots of patients to see” (work overload), “…many patients to see” (work overload). 

Related to the team: “there never is one! We always do it by ourselves...” (absent or inadequate tutoring). 

Related to the patient: “critical patients” (complex patient), “very long prescriptions with many items” (polypharmacy). 

Related to the task: “if a boss is from one segment and the other from another, or if the shift changes, all practices change” (Inadequate protocol). 

 These are some examples of the participants' speeches that we used as examples in the manuscript and that with the help of the Nvivo software and spreadsheets in excel, made into a table. We have added a note about how this table was made in line 392.

In tables 4 (old 2) and 5 (new) (line 445), and in the analysis in general, we made the "themes" clearer. That is, the themes identified and emerged with Reason's theoretical model, were sought, refined and interpreted according to the reports and subjective realities of the participants. Which we believe is relevant and novel, since we did not find similar studies in Brazil and there are important findings that differ from the current literature.

Our study followed a pattern widely used in other studies [3,4,17,18,22-29] which allowed comparison and dialogue between the studies. 

Also, the quotes do not have the annonymised details of the junior dr. My general impression is that the authors have not analysed their data in depth sufficiently for a qualitative study.

Response: Codes were used in our data analysis, however, at first, we did not put the codes in the manuscript to hinder participant identification, however, we agree with the suggestion and added in the selected reports the respective codes. (lines 254 to 255)

Response: We accept your suggestions and criticism and have improved the data analysis. We believe we have addressed all suggestions and that it has greatly improved our manuscript. Thank you

Discussion Too wordy. Needs to be summarised.

Response: We accept the suggestion

---

## [Decision Letter · Decision Letter 1]

15 Sep 2022

PONE-D-21-03374R1Prescribing errors in a brazilian teaching hospital: causes and underlying factors from the perspective of junior doctorsPLOS ONE

Dear Dr. Bonella,

Thank you for submitting your manuscript to PLOS ONE. After careful consideration, we feel that it has merit but does not fully meet PLOS ONE’s publication criteria as it currently stands. Therefore, we invite you to submit a revised version of the manuscript that addresses the points raised during the review process.

We look forward to receiving your revised manuscript.

Kind regards,

Syed Ilyas Shehnaz

Academic Editor

PLOS ONE

Reviewers' comments:

Reviewer's Responses to Questions

**Comments to the Author**

1. If the authors have adequately addressed your comments raised in a previous round of review and you feel that this manuscript is now acceptable for publication, you may indicate that here to bypass the “Comments to the Author” section, enter your conflict of interest statement in the “Confidential to Editor” section, and submit your "Accept" recommendation.

Reviewer #1: All comments have been addressed

Reviewer #3: (No Response)

2. Is the manuscript technically sound, and do the data support the conclusions?

Reviewer #1: Partly

Reviewer #3: Partly

3. Has the statistical analysis been performed appropriately and rigorously? 

Reviewer #1: N/A

Reviewer #3: N/A

4. Have the authors made all data underlying the findings in their manuscript fully available?

Reviewer #1: Yes

Reviewer #3: Yes

5. Is the manuscript presented in an intelligible fashion and written in standard English?

Reviewer #1: Yes

Reviewer #3: Yes

6. Review Comments to the Author

Reviewer #1: 1. On the style of writing – agree with Reviewer 2 that the writing is lengthy while at the same time not easy to digest the messages that the investigators are trying to put across to readers.

2. Methodology – despite reading the article, it is not apparent from the account on the timeframe during which the interview with the junior doctors was performed. Was it over a year and the number of errors committed an annual figure?

3. Due to the initial nil response rate on the investigators’ email invitation, they elected to recruited interviewees at their workplace in person or by telephone. Was it a random selection taking into account only the recruitment of doctors from as diverse a background as possible? Otherwise, it could be argued that despite their busy work schedule, only those who have a lot of grievances to air would choose to be interviewed.

4. Was the reporting of errors by recalling from memory by the junior doctors or their peers only? This is subject to recall bias, and the number might not reflect the true magnitude.

5. What about reporting by other healthcare professionals such as nurses and pharmacists?

6. There were reports of many ‘Violations’ – defined as voluntary actions in which rules are ignored, such as not evaluating the patients before copying and repeating the last prescriptions. While this situation is definitely unsatisfactory, these were forced upon the junior doctors from insufficient allowance of adequate time for them to perform the many duties that they had to perform. Is it appropriate to talk about reprimanding the doctors rather than rectifying the work conditions?

7. Many comments from the junior doctors were included in the text. They were, however, shown as disjointed messages. By such presentation to the readers, not only that there seems a risk the comments were taken out of context, sometimes the messages themselves were not entirely clear in meaning. The investigators could just describe a system through which they would categorize these comments and present the categorized results.

Reviewer #3: Thanks so much for letting review this revised manuscript. I've taken the time to read your responses to the reviewers as well as the manuscript itself.

Overall, I think the paper still drifts into quantitative presentation when you have used an entirely qualitative approach. I hope these points are swift to address.

1) Please clarify how the prescribing errors reported in Table 1 were identified. Were these self reported by practitioners, or were they identified through another aspect of the study

2) How were the 40 approached clinicians identified?

3) Table 2 requires only the demography, and not the wider profile of prescribing errors. These aren't important to the analysis

4) Table 3 is unnecessary.

5) I suggest you review the COREC checklist for reporting of qualitative research to guide your manuscript. Most of it is done but there are some aspects omitted.

6) An aspect of the qualitative approach is how you as the analysts would assign the prescribing errors in Table 3 to one of Reason's latent factors. e.g. were omissions an RBM, KBM, slip/lapse or violation?

7) I find it interesting that you've decided you're using a phenomenological paradigm, but then use grounded approaches to your data analysis. As I read your manuscript I very much see a well designed thematic analysis approach (particularly as you have used the Framework approach to data management and analysis.)

8) With regards your analysis, what is your unit of analysis?

9) As I read your results and analysis I wonder if the analysis has been somewhat superficial. You have made a good descriptive presentation of errors grouped using Reason's approach, but then have considered error-producing conditions almost as a separate entity - this isn't really the right way to consider it (and indeed, Lewis et al. wove these provoking events into the typology of errors to give deep insight into what was going on...) Have a look at your data again and see if there's a relationship between, say, team and environment and RBMs and violations.

10) Please review some of the language that has permeated into the analysis. Page 22, Line 479 onward, there is very clear potential for attribution bias in some of the direct quotes used, and how that has been interpretted. Demanding that the site of study should be closed down and criminal accusations... may be related to teh passion and hyperbole of the participant but isn't wise to feature in a journal article that your employers will read and want to follow up on. Please consider how these can be paraphrased to retain the relevance to the study outcomes, and not embarass the host site.

7. PLOS authors have the option to publish the peer review history of their article (what does this mean?). If published, this will include your full peer review and any attached files.

Reviewer #1: No

Reviewer #3: No

---

## [Author Response · Author response to Decision Letter 1]

15 Oct 2022

ANSWER TO REVIEWERS

We thank the referees and the editor for their considerations, which we have treated with care in the responses, review and rewriting of the text.

Reviewer #1: 

1) On the style of writing – agree with Reviewer 2 that the writing is lengthy while at the same time not easy to digest the messages that the investigators are trying to put across to readers. 

RESPONSE

Thank you for taking part of your time to contribute to the improvement of our manuscript.

We agree that the writing was long and we have cut some unnecessary writings. The subject matter and the reality studied by the research is probably the cause of the message not being easy to digest, however we agree with the reviewers that the text is intelligible and written in standard English. 

2) Methodology – despite reading the article, it is not apparent from the account on the timeframe during which the interview with the junior doctors was performed. Was it over a year and the number of errors committed an annual figure? 

RESPONSE

The interviews lasted 4 months, from April to July 2018.

The number of errors (105) is not an annual number, as it refers to the entire professional life reported and remembered by the interviewees. We have rewritten the text to make this more prominent.

…in the period between April and July 2018, when the interview was performed.

…throughout their entire professional life.

3)Due to the initial nil response rate on the investigators’ email invitation, they elected to recruited interviewees at their workplace in person or by telephone. Was it a random selection taking into account only the recruitment of doctors from as diverse a background as possible? Otherwise, it could be argued that despite their busy work schedule, only those who have a lot of grievances to air would choose to be interviewed.

RESPONSE

This was a random selection taking into account only the recruitment of junior doctors from a background as diverse as possible.Ex.

"As one of the aims of the study was to identify contextual causes and previous errors in different hospital environments, junior doctors of both genders and from different settings (wards, adult and pediatric intensive care units and emergency care units) were selected. The study also sought to contemplate several medical specialties".

RESPONSE

The 40 doctors approached were identified through a list provided by the residency sector.

We added to the text: "The sector responsible for the medical residency programme made available a list with the junior doctors and their specialties".

4)Was the reporting of errors by recalling from memory by the junior doctors or their peers only? This is subject to recall bias, and the number might not reflect the true magnitude.

RESPONSE

Yes, by memory. Because it is a qualitative research with investigation of the reports of the doctors themselves. This is the usual form of research. The number (105) may not reflect the magnitude of the errors, but the central point of the article is mainly to explore the content of the reports, whatever the magnitude.

We have removed table 2 as its data was not relevant to qualitative research and is likely to confuse the reader. Thank you for the alert.

On the risk of memory bias, it is discussed in the analysis as a threshold for the reader to consider.

5)What about reporting by other healthcare professionals such as nurses and pharmacists?

RESPONSE

They were not the object of this specific research, but have been the subject of others in the country and may be the subject of future research.

6)There were reports of many ‘Violations’ – defined as voluntary actions in which rules are ignored, such as not evaluating the patients before copying and repeating the last prescriptions. While this situation is definitely unsatisfactory, these were forced upon the junior doctors from insufficient allowance of adequate time for them to perform the many duties that they had to perform. Is it appropriate to talk about reprimanding the doctors rather than rectifying the work conditions?

RESPONSE

We focused on the working circumstances as a condition for the occurrence of unsafe acts throughout the text, particularly in the conclusion where we suggest measures to improve these conditions, however, we believe it is appropriate to comment on this other aspect (reprimand) as it is a reality. The focus of the study was not to investigate unsafe acts committed by negligence, imprudence, and legal and ethical aspects of medical practice, which yes, should be treated differently, however, even Reason, and several other authors discuss both aspects. We mentioned reprimanding in the discussion, for the enrichment and reflection of the reader.

7)Many comments from the junior doctors were included in the text. They were, however, shown as disjointed messages. By such presentation to the readers, not only that there seems a risk the comments were taken out of context, sometimes the messages themselves were not entirely clear in meaning. The investigators could just describe a system through which they would categorize these comments and present the categorized results.

RESPONSE

We revised the text and deleted some reports that might seem disconnected or unnecessary. The reports, which are even more numerous than those exposed in this article, were chosen and placed in the text to better illustrate the identified themes and, when analyzing them, we took due care not to decontextualise them. Citing such reports is fundamental and common in qualitative articles. There is a categorization of the results in tables 3 and 4 of the new manuscript

Reviewer #3: 

Thanks so much for letting review this revised manuscript. I've taken the time to read your responses to the reviewers as well as the manuscript itself.

Overall, I think the paper still drifts into quantitative presentation when you have used an entirely qualitative approach. I hope these points are swift to address.

RESPONSE

Thank you for taking the time to review and contribute to our manuscript. One point to note about this is that our analysis is a descriptive and phenomenological exploration, and although descriptive, it resembles quantitative research in part. But in fact it is not. However, we agree with your opinion and have removed Tables 2 and 3 as they are not relevant to our results, analysis and discussion.

1) Please clarify how the prescribing errors reported in Table 1 were identified. Were these self reported by practitioners, or were they identified through another aspect of the study.

RESPONSE

Table 1 refers to explanations on the classification and definitions used in Reason's Accident Causation model. The examples we have taken from the reports. We have now added a remark in the description of this table. "The examples were taken from the reports of the participants in this study".

In Figure 1, the prescribing errors/unsafe acts reported were identified from self report, but also with conceptual analysis of these reports through Reason's Accident Causation model. 

We added in the description of the Figure: "The active failures/unsafe acts were identified from the self-report of the participants in this research.

2) How were the 40 approached clinicians identified?

RESPONSE

We added to the text: "The sector responsible for the medical residency programme made available a list with the junior doctors and their specialties".

3) Table 2 requires only the demography, and not the wider profile of prescribing errors. These aren't important to the analysis.

RESPONSE

We agreed and removed table 2

4) Table 3 is unnecessary.

RESPONSE

We agreed and removed table 3

5) I suggest you review the COREC checklist for reporting of qualitative research to guide your manuscript. Most of it is done but there are some aspects omitted.

RESPONSE

As it was not indicated which aspects were omitted, we carried out a general review.

Table 1 Consolidated criteria for reporting qualitative studies (COREQ): 32-item checklist

Domain 1: Research team and reflexivity

Personal Characteristics

1. Interviewer/facilitator Which author/s conducted the interview or focus group?

“The interviews were conducted by the first pharmaceutical MD author G, as she has over 20 years of experience in hospital pharmacy, routinely dealing with prescription errors,... “

2. Credentials What were the researcher’s credentials? E.g. PhD, MD

 Pharmaceutical MD 

3. Occupation What was their occupation at the time of the study?

RESPONSE

We add it to the text: The author who conducted all the interviews worked at the hospital pharmacy at the study site but was not known or acquainted with the participants before starting the study.

4. Gender Was the researcher male or female? 

RESPONSE

She has over 20 years of experience in hospital pharmacy,...

5. Experience and training What experience or training did the researcher have?

RESPONSE

“..she has over 20 years of experience in hospital pharmacy, routinely dealing with prescription errors,...”

Relationship with participants “…was not known or familiar with the participants before starting the study”.

6. Relationship established Was a relationship established prior to study commencement?… 

RESPONSE

was not known or familiar with the participants before starting the study.

7. Participant knowledge of the interviewer

What did the participants know about the researcher? e.g. personal goals, reasons for doing the research. 

RESPONSE

… was not known or familiar with the participants before starting the study.

We added: Reasons and interests were scientific in order to understand the underlying aspects of prescribing errors and were presented to the participant before starting the interview.

8. Interviewer characteristics What characteristics were reported about the interviewer/facilitator? e.g. Bias, assumptions, reasons and interests in the research topic

RESPONSE

The author who conducted all the interviews worked at the hospital pharmacy at the study site but was not known or acquainted with the participants before starting the study. The medical prescriptions are usually printed directly in the pharmacy sector and when a problem is detected, the pharmaceutical team contacts the doctor by phone for clarification. No member of the study taught, supervised, or had any authority over participants in the study. 

The interviewer was fully aware of her potential assumptions during the interviews, taking care that these did not interfere with the accounts, she sought not to deviate from the questions contained in the interview guide, ensured that all data could be reviewed and discussed by the entire research team and that disagreements could be discussed and resolved by consensus ensuring neutrality and objectivity to the study. 

We added: Reasons and interests were scientific in order to understand the underlying aspects of prescribing errors.

Domain 2: study design

Theoretical framework

9. Methodological orientation and Theory

What methodological orientation was stated to underpin the study? e.g. grounded theory, discourse analysis, ethnography, phenomenology, content analysis

RESPONSE

We conducted a qualitative, descriptive exploratory study, collecting data through individual interviews. Focusing on the subjective and personal dimension of human error, we adopted a constructivist paradigm, assuming that there are multiple and distinct subjective realities (perspectives) [31]. Within this paradigm, we adopted Reason’s Accident Causation model [21] for the concept-based exploration of themes . The epistemic dimension was phenomenological [31], starting from the way reality appears to the participants themselves, taking them as autonomous subjects of medical practice.

Participant selection

10. Sampling How were participants selected? e.g. purposive, convenience, consecutive, snowball

RESPONSE

Invitation to participate in the study was initially done through an email to all (632) junior doctors enrolled in the Medical Residency Program, however there were no responses. The sector responsible for the medical residency program provided a list of junior doctors and their specialties.

11. Method of approach How were participants approached? e.g. face-to-face, telephone, mail, email

RESPONSE

Thus 40 participants were individually recruited in their workplace, either in person or by telephone. As one of the aims of the study was to identify contextual causes and previous errors in different hospital environments, junior doctors of both genders and from different settings (wards, adult and pediatric intensive care units and emergency care units) were selected. The study also sought to contemplate several medical specialties. 

12. Sample size How many participants were in the study?

RESPONSE

40 participants

13. Non-participation How many people refused to participate or dropped out? Reasons?

RESPONSE

None of the recruited participants refused to participate in the study, however, six did not participate due to their impossibility to schedule appointments with the interviewer, which, according to them, was due to their residency schedule and work overload. Theoretical saturation was obtained with 34 interviews [32]. 

Setting

14. Setting of data collection Where was the data collected? e.g. home, clinic, workplace

RESPONSE

Were individually recruited in their workplace,

15. Presence of non-participants Was anyone else present besides the participants and researchers?

RESPONSE

There was no one else present during the interviews. 

“The interviews were conducted in a private place,”

16. Description of sample What are the important characteristics of the sample? e.g. demographic data, date

RESPONSE

The sampling universe consisted of doctors from the residency program who were training at the hospital in the period between April and July 2018 during which the interview was performed. First, second and third-year residents were included in the study, those with over three years and those on vacation or on sick leave during the study period were excluded. 

Data collection

17. Interview guide Were questions, prompts, guides provided by the authors? Was it pilot tested?

RESPONSE

An interview topic guide was used with questions related to the “planning” and “execution” of the prescription (S1 and S2), and was an adaptation of the one used in Lewis et al. [26], 

18. Repeat interviews Were repeat interviews carried out? If yes, how many?

RESPONSE

No. We added to the manuscript text: "... and the interviews were not repeated and the transcripts were not returned to the participant for comment or correction."

19. Audio/visual recording Did the research use audio or visual recording to collect the data?

RESPONSE

They were audio-recorded 

20. Field notes Were field notes made during and/or after the interview or focus group?

RESPONSE

Yes. We added to the manuscript text, "Field notes were taken, however interviews were not repeated and transcripts were not returned to the participant for comment or correction."

21. Duration What was the duration of the interviews or focus group?

RESPONSE

The interviews lasted between 15 and 35 minutes 

22. Data saturation Was data saturation discussed?

RESPONSE

Theoretical saturation was obtained with 34 interviews [32]. 

23. Transcripts returned Were transcripts returned to participants for comment and/or correction?

RESPONSE

No. We added to the manuscript text, "Field notes were taken, however interviews were not repeated and transcripts were not returned to the participant for comment or correction."

Domain 3: analysis and findings

Data analysis

24. Number of data coders How many data coders coded the data?

RESPONSE

The codes and sub-codes were pre-established according to Reason's Accident Causation model. They are detailed in Figure 1. 

25. Description of the coding tree Did authors provide a description of the coding tree?

RESPONSE

The Coding tree is represented in Figure 1.

The development of the analytic framework and a categorization and coding according to “Reason's Accident Causation model” were carefully elaborated by G. This model was the most commonly used theoretical model when considering prescribing errors [3,4,17,18,22-29]. To ensure higher accuracy, the application of this framework and the mapping of the data were performed independently by G and L and the differences were settled by consensus. Tables and organizational charts were constructed to facilitate understanding and interpretation. 

The computer software program NVivo© [35] was used to assist in the organization of the data. Codes and sub-codes to qualify the 34 cases were previously defined and used during the analysis process to identify potential themes. Prescribing errors were then counted and named according to the taxonomy in Otero-López et al. [36]. 

26. Derivation of themes Were themes identified in advance or derived from the data?

RESPONSE

Codes and subcodes to qualify the 34 cases were previously defined and used during the analysis process to identify potential themes. 

27. Software What software, if applicable, was used to manage the data?

RESPONSE

The computer software program NVivo© [35] was used to assist in the organization of the data.

28. Participant checking Did participants provide feedback on the findings?

RESPONSE

yes

Reporting

29. Quotations presented Were participant quotations presented to illustrate the themes / findings? Was each quotation identified? e.g. participant number

RESPONSE

For a better qualitative understanding of the data, literal reports were selected to illustrate the main results. The quotes were referenced using numbers representing the participant (P) code, eg., P 01, P02, etc.

30. Data and findings consistent Was there consistency between the data presented and the findings? RESPONSE

Yes

31. Clarity of major themes Were major themes clearly presented in the findings?

RESPONSE

The themes were detailed in tables 2 and 3. Those that stood out the most and those that stood out the least were discussed in the text that followed the tables.

32. Clarity of minor themes Is there a description of diverse cases or discussion of minor themes?

RESPONSE

Minor themes were also discussed in the text that followed the tables.

6) An aspect of the qualitative approach is how you as the analysts would assign the prescribing errors in Table 3 to one of Reason's latent factors. e.g. were omissions an RBM, KBM, slip/lapse or violation? 

RESPONSE

Omissions (taxonomy of Otero-Lópes et al [36]) could have been caused by either RBM or, KBM, slip/lapse or violation (all of which are unsafe acts). However in our research, Omissions were mostly attributed to violations. Example taken from the manuscript:

“Many violations happened this way. Junior doctors made a copy of the prescriptions, planning to return to the system later to check and update it, but never did (that is a violation/unsafe act ("Reason 's Accident Causation model" [21]). The most common errors related to violations were drugs being unnecessarily prescribed (Unnecessary medication/table 3 - This is an error type according to the taxonomy of Otero-Lópes et al [36]) and medication omission (Medication Omission table 3 - This is an error type according to the taxonomy of Otero-Lópes et al [36].). Copying prescriptions from the previous day without properly checking them may perpetuate errors for many days”.

Note: What is in red does not appear in the text of the manuscript, we just flag it here for your better understanding.

In the results and analysis we did this, for all reports we related Types of errors (Otero-Lópes et al [36] taxonomy) with the Unsafe Acts/Active Failures (Reason 's model" [21]) and identified the Themes (Table 2- Themes identified from the reports as error-producing conditions and Table 3 - Themes identified from the reports as latent conditions). 

Medication omission, wrong dose, wrong medication, wrong route of administration and wrong patient were also mostly related to Slips (unsafe acts) and had as error-producing conditions the "electronic system" used at the study site (Table 2 - Themes identified from the reports as error-producing conditions).

See this in the text below, taken from the manuscript, where we analyze this relationship:

“Most slips (“Reason's Accident Causation model”[21]) were related to medication omission and wrong dose, wrong medication, wrong route of administration and wrong patient (Otero-Lópes et. al. [36]). Due to name similarity between the drugs and their alphabetical proximity in the prescribing electronic system (That is an error-producing condition - Table 2 - Themes identified from the reports as error-producing conditions ), the interviewee selected the wrong medication (Otero-Lópes et. al. [36])”. 

Note: What is in red does not appear in the text of the manuscript, we just flag it here for your better understanding.

In summary, Reason [21] classifies the active failures/unsafe acts as "errors" ( RBM, KBM, slip/lapse) and "violations"; Otero-Lópes et. al. [36] classify the "error types" using a specific taxonomy without any mention to the unsafe act that caused such error. These two analyzed aspects provided support for the identification of the THEMES (Table 2 and 3). 

7) I find it interesting that you've decided you're using a phenomenological paradigm, but then use grounded approaches to your data analysis. As I read your manuscript I very much see a well designed thematic analysis approach (particularly as you have used the Framework approach to data management and analysis.)

RESPONSE

The overall approach is phenomenological as it was about understanding the subjectivity of the respondents, but the data management was with the framework and the data analysis approach was thematic. A multi-factorial type of methodology.

8) With regards your analysis, what is your unit of analysis?

RESPONSE

Reason's categorisation of medical errors. Over the course of the research, the central unit was violations as the highlight of the results.

9) As I read your results and analysis I wonder if the analysis has been somewhat superficial. You have made a good descriptive presentation of errors grouped using Reason's approach, but then have considered error-producing conditions almost as a separate entity - this isn't really the right way to consider it (and indeed, Lewis et al. wove these provoking events into the typology of errors to give deep insight into what was going on...) Have a look at your data again and see if there's a relationship between, say, team and environment and RBMs and violations.

RESPONSE

We thank you for your alert for the correction. We corrected the text in order to deepen our analysis and leave the relationship between unsafe acts and their error-producing conditions in greater focus, as it was indeed our intention.

11) Please review some of the language that has permeated into the analysis. Page 22, Line 479 onward, there is very clear potential for attribution bias in some of the direct quotes used, and how that has been interpretted. Demanding that the site of study should be closed down and criminal accusations... may be related to teh passion and hyperbole of the participant but isn't wise to feature in a journal article that your employers will read and want to follow up on. Please consider how these can be paraphrased to retain the relevance to the study outcomes, and not embarass the host site..

RESPONSE

We agree. We modified the text. 

Thank you for your considerations and corrections.

---

## [Decision Letter · Decision Letter 2]

6 Mar 2023

PONE-D-21-03374R2Prescribing errors in a Brazilian teaching hospital: causes and underlying factors from the perspective of junior doctors

Dear Dr. Bonella,

Thank you for submitting your manuscript to PLOS ONE. After careful consideration, we feel that it has merit but does not fully meet PLOS ONE’s publication criteria as it currently stands. Therefore, we invite you to submit a revised version of the manuscript that addresses the points raised during the review process.

We look forward to receiving your revised manuscript.

Kind regards,

Deema Jaber, Ph.D.

Academic Editor

PLOS ONE

Reviewers' comments:

Reviewer's Responses to Questions

**Comments to the Author**

****

1. If the authors have adequately addressed your comments raised in a previous round of review and you feel that this manuscript is now acceptable for publication, you may indicate that here to bypass the “Comments to the Author” section, enter your conflict of interest statement in the “Confidential to Editor” section, and submit your "Accept" recommendation.

Reviewer #3: All comments have been addressed

2. Is the manuscript technically sound, and do the data support the conclusions?

Reviewer #3: Yes

3. Has the statistical analysis been performed appropriately and rigorously? 

Reviewer #3: N/A

4. Have the authors made all data underlying the findings in their manuscript fully available?

Reviewer #3: No

5. Is the manuscript presented in an intelligible fashion and written in standard English?

Reviewer #3: Yes

6. Review Comments to the Author

Thank you for inviting me to review this manuscript which describes the prescribing errors in a brazilian teaching hospital.

Overall, I find the manuscript somewhat confusing. The authors have used a qualitative approach to explore the causes of medication prescribing errors among junior doctors. However, the results presented are very quantitative. Table 1 and 2. The manuscript is also very wordy. The authors should consider cutting down on the number of words, and make their writing more concise.

Introduction

I like the model (i.e. Figure 1) which is used to explain the conceptual framework of this study. However, the authors tend to repeat what is in the figure as text. Please avoid this. Conceptual framework should be summarised, and jusitification for selecting this theory should be provided The authors should provide a stronger justification on why this work is needed. Note, that any abbreviations used in the figure (e.g. KBM) should be explained as a footnote. Aim should be changed to explore .... (rather than investigate - which is very quantitative)

Methods

Should use more subheadings in this section to aid reading. How was the topic guide developed? should include the questions asked as a table. Where's the rigour and reflexivity sections?

Results

Lines 235-242 should be presented as a table to decrease the number of words.

Table 2: how did the authors collect this data? Did they use a survey form? Very quantitative results

For a qualitative paper, I do not see what themes emerged from the participants. I only see a repetition of the conceptual framework (i.e. results are presented according to the conceptual framework). As such, I don't think the authors actually explored why the junior doctors made the prescribing errors. Also, the quotes do not have the annonymised details of the junior dr. My general impression is that the authors have not analysed their data in depth sufficiently for a qualitative study.

Discussion

Too wordy. Needs to be summarised.

Reviewer #3: Thanks for taking the time to respond to my comments. I am satisfied with your responses. Well done.

7. PLOS authors have the option to publish the peer review history of their article (what does this mean?). If published, this will include your full peer review and any attached files.

---

## [Author Response · Author response to Decision Letter 2]

16 Mar 2023

RESPONSE TO REVIEWERS

Reviewer #3: Thanks for taking the time to respond to my comments. I am satisfied with your responses. Well done.

Response: Thank you, we have clarified to the editor that you have already approved and were satisfied with our responses

---

## [Editor Report · Decision Letter 3]

23 Mar 2023

Prescribing errors in a Brazilian teaching hospital: causes and underlying factors from the perspective of junior doctors

PONE-D-21-03374R3

Dear Dr. Bonella,

We’re pleased to inform you that your manuscript has been judged scientifically suitable for publication and will be formally accepted for publication once it meets all outstanding technical requirements.

Kind regards,

Deema Jaber, Ph.D.

Academic Editor

PLOS ONE
---

## [Editor Report · Acceptance letter]

28 Mar 2023

PONE-D-21-03374R3 

Prescribing errors in a Brazilian teaching hospital: causes and underlying factors from the perspective of junior doctors 

Dear Dr. Bonella:

I'm pleased to inform you that your manuscript has been deemed suitable for publication in PLOS ONE. Congratulations! Your manuscript is now with our production department. 

Kind regards, 

on behalf of

Dr. Deema Jaber 

Academic Editor

PLOS ONE